# Laser-Assisted Direct Grafting of Poly(ethyleneimine) on Poly(methyl methacrylate)

**DOI:** 10.3390/polym14102041

**Published:** 2022-05-17

**Authors:** Hyeyoung Park, Martin Wiesing, Philipp Zimmermann, Andreas Janke, Simona Schwarz, Jürgen Nagel

**Affiliations:** 1Leibniz-Institut für Polymerforschung Dresden e.V., Hohe Straße 6, 01069 Dresden, Germany; park-hyeyoung@ipfdd.de (H.P.); zimmermann-philipp@ipfdd.de (P.Z.); andy@ipfdd.de (A.J.); simsch@ipfdd.de (S.S.); 2Fraunhofer-Institut für Fertigungstechnik und Angewandte Materialforschung (IFAM), Wiener Straße 12, 28359 Bremen, Germany; martin.wiesing@ifam.fraunhofer.de

**Keywords:** polymer, grafting, surface modification, photothermal reaction, laser

## Abstract

Demand for direct chemical modification of functional material on a surface is increasing in various fields. A new approach for a functionalized surface is investigated by applying a conventional laser in order to generate chemical activation by photothermal energy. Poly(ethyleneimine) (PEI), with a high density of amino groups, is chemically grafted on poly(methyl methacrylate) (PMMA) by irradiation of a CO_2_ laser (10.6 μm). Laser parameters such as power, scan rate, and focal length are observed to play an important role in order to introduce effective photothermal energy for the chemical reaction between PEI and PMMA. By optimization of laser parameters, the amide compound is produced as a result of the reaction of amine from PEI and the ester of PMMA successfully. The PMMA surface modified with PEI is analyzed by XPS and TOF-SIMS to identify the functional groups. Furthermore, the surface is characterized in terms of wettability, adhesion force, and surface charge for various applications. Finally, reaction with dye and metal on the amine-terminated PMMA shows promising results in supplying a selective and reliable functional substrate.

## 1. Introduction

Polymers have become essential components and are ubiquitous in everyday life, not only in industrial applications, due to their broad spectrum of properties [1,2] coupled with their low cost. Among them, poly(methyl methacrylate) (PMMA) is a transparent thermoplastic and has been used in a wide range of fields, such as the substitution of glass, medical technology, and daily necessities. The properties of PMMA can be modulated by compositions of monomers, molecular weight, additives, and processes to meet the purpose of the application. However, there are critical issues in surface adhesion because of its low surface energy, as with other polymeric materials. This makes further applications such as coating, biocompatibility, and metallization difficult. Surface treatment can improve the surface properties according to the purpose. Physical treatments such as plasma [3] and UV light [4] have been used to improve the adhesion of the PMMA surface by changing the roughness and breaking the chemical bonds [5]. Chemical surface modifications have been used in direct reactions with given chemicals having various functional moieties by covalent bonds, using heat for activation [6]. Most chemical reactions and processes can be improved if microwave irradiation is used instead of a traditional thermal heat source. Chemicals directly absorb microwave energy, transfer it typically to molecule vibrations, and dissipate it finally to heat. Early on, in 1967, Williams reported that microwave heating could accelerate some chemical reactions [7,8]. Since then, the microwave has been widely applied in various chemical reactions, including synthesis, solvent-free reactions [9], and the use of catalysts [10].

Over the last several years, surface modification has grown to become one of the most studied methods because of its simplicity, robustness, and numerous applications by engineering the surface’s physical, chemical, or biological characteristics, such as surface charge [11], hydrophilicity [12], roughness [13], biocompatibility [14], and reactivity [15]. Surface engineering techniques are being used in the automotive, aerospace, electronics, biomedical, and construction industries, and so on. Almost all types of materials, including metals, ceramics, polymers, and composites, can be modified by surface engineering. There have been various techniques employed, such as different kinds of plasma treatments, ion or electron beam irradiation, and also laser irradiation. Laser irradiation was introduced in order to alter the chemical and physical properties of the surface without affecting desirable bulk properties. A condition for the application of laser irradiation is the absorption by the polymer. If optical laser light is employed, an absorber has to be added to the polymer matrix. The high photon energy of ultraviolet light results in bond scissoring in the polymer, leading to less control over the type of compounds formed on the surface. Irradiation with infrared light is directly absorbed by the polymer, which initializes bond vibrations and dissipation into heat. However, the typically high doses applied result in thermal decomposition and ablation of the polymer surface layer, and, thus, less control over the chemical composition on the polymer surface. As a result, laser irradiation of polymers is typically used for material ablation. Reviews to the effect of laser irradiation on polymer surfaces have already been published [16,17]. Examples are laser graving, marking, as well as a process for preparing polymer surfaces for chemical plating [18]. No laser process is yet published that exploits the laser energy to graft polymers by direct coupling reactions to a PMMA substrate.

In this paper, we introduce the laser-assisted direct grafting of a functional polymer on PMMA. The laser energy is only used to provide the activation energy for the grafting reaction, which includes transformation of the reactants from solid to molten state, but only in a very thin surface layer on the substrate. Poly(ethyleneimine) (PEI), which is a sterically branched polymer with large numbers of amino groups, is applied as a modifier in order to demonstrate the drastic changes after modification on PMMA. An aminolysis reaction of PEI on PMMA, as shown in Figure 1, is accomplished by photo-thermal heating under control of the laser parameters. The critical laser parameters for the reaction are presented and the PEI-grafted PMMA surface is characterized to confirm the modification. Finally, the applications of the PEI-grafted PMMA are suggested and discussed.

## 2. Experimental Section

### 2.1. Materials and Equipment

Two kinds of hyperbranched PEI with weight-average molar mass 25,000 g/mol (25K PEI, SKU No 408727) and 750,000 g/mol (750K PEI, SKU No 181978) were purchased from Sigma-Aldrich. PMMA (Plexiglas ^®^GS transparent, 3 mm thickness) plates supplied by Nordic Panel GmbH were used as substrates. All commercial solvents used for preparations in the next section were analytical grade and used without further purification. Absolute ethanol was purchased from Merck. Isopropanol (99.8%, IPA) was purchased from Fisher Chemical. Deionized purified (D.I.) water used in this study was obtained from the Milli-Q system (Millipore). Eosin Y dye from Sigma-Aldrich (99%) was used as an indirect optical indicator for binding of PEI on PMMA. For the metal chelation with PEI grafted on PMMA, Copper (II) acetate was purchased from Sigma-Aldrich (Taufkirchen, Germany).

The experiments were carried out using a commercial CO_2_ laser system (V2000, Novograv e.K., Weilerbach, Germany) with a wavelength of 10.6 μm, a maximum output power of 30 W, a maximum scan rate of 1524 mm/s, and 500 PPI (pulses per inch). The output laser beam with a Gaussian transverse intensity profile was focused onto the substrates by a lens with a focal length of 38.1 mm. The samples were placed on a honeycomb stage, which was movable in the Z direction. During the laser irradiation with a laser head, moveable in X and Y directions, N_2_ gas was flushed into the sample at the same time and the atmosphere was exhausted. CorelDRAW Graphics software controlled the laser parameters, such as power, scan rate, and pulse density.

### 2.2. Preparation

Solutions of PEIs were prepared to be 1 wt.% of concentration using ethanol and IPA as solvents. These solvents and concentrations were selected due to previous promising results for spin coating on a PMMA substrate cleaned by soaking in ethanol for 1 min and rinsed thoroughly with MilliQ water. PEI layers were coated with prepared solutions on the cleaned PMMA substrates by a spin coater (Spin 150, SPS-Europe GmbH, Ingolstadt, Germany) with 2000 rpm for 30 sec at room temperature. The indicator, Eosin Y, dissolved in the mixture of ethanol and water (EtOH: H_2_O = 2:8) with a concentration of 1 mM, was used to indirectly confirm the PEI grafting after laser irradiation. Saturated Copper (II) acetate solution (72 g/L, in water) was prepared for the formation of a Cu complex on PEI-grafted PMMA.

### 2.3. Extraction and Reactions

After the laser irradiation, unreacted PEI was removed and cleaned carefully with ethanol and water. This process is called the extraction of PEI. The PEI-grafted PMMA substrate has an amine-functionalized surface with good reactivity to ions, metals, and other chemicals. To investigate ionic bonds, the amine-functionalized PMMA was immersed in 1 mM of Eosin Y solution for 10 min at room temperature and then rinsed with ethanol and water several times. For the metal chelation, the PEI-grafted PMMA was immersed in the saturated Copper (II) acetate solution for 10 min at room temperature and then rinsed with water.

### 2.4. Laser Irradiation

In order to find optimum reaction conditions between PEI and PMMA, a new experimental drawing was produced by CorelDRAW. In this paper, rectangular patterns with a gray gradient from white to black (called gray wedge) were designed and the gray level corresponds to the laser power (0 to 100% of the maximum irradiation, Figure 2a). The laser parameters, such as different power, scan rate, and working distance (Z position), were controlled by these patterns. Six patterns were irradiated by the laser to the PEI-coated PMMA substrate with different laser scan rates, 100, 80, 60, 40, and 20% of the maximum rate (1524 mm/s). The laser beam sizes were varied by the Z position of the stage with 0, 10, 15, and 20 mm.

The grafted area was visualized by Eosin Y (Figure 2b) adsorption to the PEI-grafted PMMA surface. The selective color change from orange to pink after extraction of unbound PEI molecules from the PMMA surface pointed to the grafting of PEI.

### 2.5. Characterization

In order to identify the functional groups before and after modification of PEI on PMMA, X-ray photoelectron spectroscopy (XPS) spectra were obtained by using the Kratos Axis Ultra XPS instrument (Kratos Analytical, Manchester, UK) using a monochromatized AlKα (1486.6 eV), neutralization by low-energy electrons, and detection of electrons along the surface normal (0°). The pass energies were set to 150 eV (survey spectra) and 20 eV (core level spectra), where the latter corresponds to a resolution of 0.6 eV of the Ag3d5/2 line. The spectrometer energy scale was calibrated by shifting C1s to 285.0 eV. Calibration of the C1s spectrum of neat PEI was referenced to 285.6 eV according to Beamson and Briggs [19]. Spectral fitting was done by using CasaXPS (Casa Software Ltd., Teignmouth, UK) [20]. 

Time-of-flight secondary ion mass spectrometry (TOF-SIMS) data were acquired using a TOF-SIMS IV (ION-TOF GmbH, Münster, German) with a 30 keV Bi + ion source. Data were taken over a mass range up to 800 *m*/*z* on 500 µm × 500 μm at 512 pixel × 512 pixel (resolution around 2–3 μm). Characteristic fragments in positive mode were seen, such as the methacryloyl ion at *m*/*z* 69, among others, at *m*/*z* 85, 109, 125, and 139. Assignment of fragments was alleviated by principal component (PCA) analysis of the distribution images, because a heterogeneous surface chemistry was observed on some samples after surface modification. The spectral interpretation was focused on identifying characteristic ions of the PEI on the surface. 

XPS and TOF-SIMS samples were prepared with cleaned bare PMMA, PEI-coated PMMA with 2 wt.% of 25K PEI in ethanol, and PEI-grafted PMMA, which was obtained with optimum laser irradiation (25.5 W, 1219.2 m/s at Z = 20 mm), and we extracted unbounded PEI on the PMMA by rinsing with ethanol and D.I. water. 

The surface morphology was analyzed by scanning electron microscopy (SEM) using a Phenom microscope (FEI Company, Fremont, CA, USA) as well as an Ultra Plus Gemini microscope (Carl Zeiss SMT GmbH, Jena, Germany). For topology characterization, SE detection, and element mapping and identification, an EDX detector, XFlash 5060 F (Bruker Corporation, Billerica, MA, USA), was used. 

Adhesion force measurements were carried out using an atomic force microscope (AFM), Dimension FastScan (Bruker-Nano, Billerica, MA USA). Silicon cantilevers, RTESPA-150-30 (Bruker-Nano, USA), were used, with a measured spring constant of 5.06 N/m. AFM topographic images were obtained with the tip velocity of 1 um/s in 20 μm × 20 μm size in peak force tapping mode. The film thickness was calculated from the image by measuring the gap between the highest and lowest. The adhesion force curves were measured in a 3 × 3 array with a distance of 200 nm between the points and analyzed by the NanoScope Analysis software v. 1.9 (Bruker-Nano, Billerica, MA, USA). All the adhesion forces were measured after baseline correction. For AFM measurements, the bare PMMA sample was cleaned with D.I. water and ethanol, and the PEI-coated PMMA was prepared with 750K PEI in IPA of 1 wt.% concentration on the cleaned PMMA substrates by spin coating at 2000 rpm for 30 s at room temperature. The PEI-grafted PMMA sample was prepared following the optimum laser parameters (21 W, 914.4 m/s at Z = 20 mm) for laser irradiation on the PEI-coated PMMA, which was coated with 1 wt.% of 750K PEI, and the extraction process was performed after irradiation.

The electrokinetic measurements were performed using an Electrokinetic Analyser (EKA) (Anton Paar KG, Graz, Austria), by measuring the streaming potential between two electrodes. Two glass substrates were tightly mounted between the two Ag/AgCl electrodes in the measuring cell. A 10−3M aqueous solution of KCl was used as a test solution for all measurements. The electrolyte concentration was kept nearly constant while pH was adjusted by adding HCl or KOH to the KCl electrolyte solution in order to perform the measurements in the pH range from 3 up to 10. The zeta potential (ζ) values were calculated according to the Smoluchowski equation [21,22] from the experimentally determined streaming potentials. The samples were prepared with cleaned bare PMMA, and 750 K PEI-grafted PMMA, which was obtained with optimum laser irradiation (21 W, 914.4 m/s at Z = 20 mm), and we extracted unbounded PEI on the PMMA by rinsing with ethanol and D.I. water.

The wettability of each surface was determined by measurement of the static water and 0.1 M HCl contact angle using a Krüss G2 contact angle goniometer (Krüss GmbH, Hamburg, Germany). The analysis was carried out by the software Drop Shape Analysis (Krüss GmbH, Hamburg, Germany). All samples were prepared in the same manner as the ones for zeta potential measurements. UV–Vis optical spectra were measured with a Cary 60 spectrometer (Agilent, Santa Clara, CA, USA).

## 3. Results and Discussion

### 3.1. Effect of Energy Density for Grafting of PEI on PMMA

The chemical bonding of amine from the PEI with the ester group on the PMMA substrate depended on the energy density of the CO_2_ laser. The thickness of the PEI layer spin-coated on PMMA was measured by AFM measurement as 125 nm and 179 nm for 25K and 750K PEI, respectively. In principle, the PEI and PMMA molecules absorb light of a wavelength of 10.6 μm (943.4 cm^−1^) of the laser, which corresponds to the wavenumbers for bending, vibration, and stretching of organic material components. As a result of laser irradiation, the light energy is transformed into photo-thermal heating, which becomes the activation energy for chemical reactions. Aminolysis reactions are affected mainly by temperature, time, and pH value [23]. It has been reported that the reaction between PMMA and PEI in NMP solvent happens at 110 °C for 12 h [24]. Therefore, the photo-thermal heating induced by laser treatment onto the PMMA substrate should be carefully controlled for the aminolysis reaction. The critical point is how much energy density is needed for chemical bonding in a solid state. To find the optimum energy density for the grafting of PEI on a PMMA substrate, laser parameters were investigated, as mentioned in the Experimental Section; see Figure 2a. When a focused beam of laser is irradiated on a PEI-coated PMMA surface, photochemical ablation, which is a type of decomposition combined with both melting and vaporization of PMMA and its fractions, occurs due to the breaking of covalent bonds in polymer chains by the absorption of the photon energy. Several studies focused on the mechanism of the ablation of materials via laser [25,26,27]. As a result, the irradiated areas of the PEI-coated PMMA were ablated. The depth of features created by focused laser ablation was increased by increasing the energy density and by decreasing the scan rate. When focused, the intensity of the laser was too strong for the surface reaction, which resulted in ablation. In order to avoid ablation by controlling the energy density, the Gaussian shape of the beam should be adjusted by changing the sample position (Z).

The laser beam diameter increases with the distance between the focal point and material surface plane, as shown in Figure 3a,b. In Figure 3b, the peak intensities were calculated by the beam diameters obtained by simple mathematics according to the beam size (approximately 0.206 mm) at each Z position [28]. The beam diameter corresponds to the beam width, where the intensity drops to 1/e^2^ of the peak intensity [29]. By increasing the Z position, the laser beam diameter became larger, with lower peak intensity. To investigate the optimum condition for carrying out the aminolysis reaction, the PEI-coated PMMA substrates were irradiated by different laser powers and scan speeds, with variations in the Z position, which are shown in Figure 4a–e. After laser treatment, the PEI-grafted PMMA regions were visualized by Eosin Y absorption after the extraction process of the unbounded PEI from the PMMA surface. With high energy density, the ablation phenomenon was dominant, so that the aminolysis reaction was not observed, as shown in Figure 4a,b. Upon increasing the Z position, selective absorptions of Eosin Y were visualized as a pink color only on the PEI-grafted PMMA area. 

The required energy for the PEI grafting on the PMMA substrate has to be increased as a function of the distance (Z). The area of pink where the Eosin Y reacted was shifted up to the high power from 30~50% to 50~70% and 60~80%, respectively as the Z increased from 10 mm to 15 mm and 20 mm at 60% of scan speed. As the Z increases, the beam diameter increases, and the energy density of the laser decreases; therefore, more energy for the reaction is required. Furthermore, the Z plays a critical role in the chemical reaction. In focused irradiation, the heating effect is mostly imparted to the material and it presents decomposition, which includes melting, gasification, and evaporation, as shown in Figure 4a,b. It is necessary to provide maximum heat while not inducing the temperature to exceed 160 °C, where the material starts to soften. The temperature induced by the laser can be modulated properly by broadening the laser beam according to the Z position. As a result, the optimum laser condition for PEI modification on the PMMA substrate was achieved by controlling the laser power, scan speed, and Z by calculating how much energy is required for PEI grafting on the PMMA surface. Presumably, the temperature in the irradiated area stays between the annealing and decomposition temperatures of PMMA. 

#### 3.1.1. X-ray Photoelectron Spectroscopy (XPS) and Time-of-Flight Secondary Ion Mass Spectrometry (TOF-SIMS)

X-ray photoelectron spectroscopy (XPS) was performed in order to identify the elemental composition of the surface layer, as well as the nature of the coupling between PMMA and PEI (25K PEI). The highly resolved spectrum (Figure 5) revealed the elements C, O, and N, as shown in Table 1. Whereas C and O stem obviously from the PMMA substrate, the N signal pointed to a considerable amount of bonded PEI on the surface layer. This means that the PEI layer grafted on PMMA can be considered as a monolayer range after removing the physically adsorbed PEI. Furthermore, the thickness of the PEI-modified layer was calculated to be approximately 0.4 nm, and the component was homogeneous according to and in the framework of the XPS analysis.

To elucidate the chemical interactions between C in PMMA and N in PEI, highly resolved spectra of the appropriate spectral regions were recorded. Figure 5(a) shows the typical C1s spectrum of the bare PMMA substrate, which was mathematically described using four individual chemical components, i.e., the CC and CH groups at 285.0 eV, (C*COO) at 285.7 eV, C-O of the ester at 286.9 eV, and the carboxylate COO at 288.9 eV [30]. It is noted that the widths of all components have been fixed to be equal. The area of the C*COO component has also been set to the value found for the parent COO component. The results show that the overall spectrum is very consistent with PMMA spectra from the literature [27]. The analysis of the C1s and N1s spectra of the PEI-coated PMMA shows a single component peak in the C1s, which has been set to 285.5 eV according to the literature [27]. The binding energy found for the N1s was consequently found at 398.9 eV, which is again in agreement with the literature and demonstrates that PEI-related amine groups can be well differentiated from most other organic N compounds, which typically show N1s peaks around 400 or 401.5 eV (ammonium ion). The oxidation of PEI during laser treatment could be excluded due to the nitrogen purging during the process. After laser treatment on the PEI-coated PMMA, the XPS spectrum of C1s showed no significant change (Figure 5c). However, a new component was detected at 400.0 eV, which is typical for a broad range of organic N compounds, e.g., non-PEI amines, amides, and nitriles, as shown in Figure 5d [31]. It is thus proposed that the immobilization of PEI is a consequence of the formation of amide bonds with the PEI layer on the PMMA surface. This covalent bonding is the apparent reason for the high adhesion and resistance against extraction in ethanol. 

The surface chemical characterization was complemented by a ToF-SIMS analysis of the PEI-coated and grafted PMMA surfaces. However, the comparison of the spectra did not reveal any new characteristic fragment signals upon immobilization. This finding is in agreement with the fact that amide compounds generally do often not yield characteristic fragments other than CNO- and also PEI lacks clear fragments [32]. The analysis of the spectra was thus limited to the comparison of the CN-, CNO-, and NH- signal intensities, as shown in Table 2. The results show that there is a tendency of the CNO- fragment intensity to increase relative to the CN- intensity from 0.39 to 0.44. This finding has to interpreted with great care, because ToF-SIMS is a predominantly qualitative technique and prone to matrix effects. However, these results may yield a subtle corroboration of the hypothesized presence of amide linkages between PEI and PMMA.

#### 3.1.2. Adhesion Force, AFM

Adhesion force was studied for the investigation of PEI modification by AFM. In Figure 6, representative force–distance curves after baseline corrections for bare PMMA, PEI-coated PMMA, and PEI-grafted PMMA are shown. During extension and retraction of the AFM tip, the force was measured as a function of distance. The point of contact can be assumed to be the point at which the force is zero. After contact, this force is considered as the interaction force between the AFM tip and the surface. During retraction after maximum extension, the force decreases and reaches negative values due to adhesion. After the force reaches the minimum (maximum adhesion force) [33,34,35] (Figure 6(a)–(c)), the complete detachment of the tip from the surface occurs until the force reaches zero. The rupture length is defined as the length from the force of zero, which represents the starting point for detachment, to the complete detachment of the tip (Figure 6(a’)–(c’)). The dashed curve is the retraction force of the AFM tip from the bare PMMA surface, showing that the maximum adhesion force is 40.8 nN (Figure 6(a)) and the rupture length is 6.1 nm (Figure 6(a’), Table 2). Compared to bare PMMA, the PEI-coated PMMA shows higher adhesion force and a longer rupture length. This correlates with the presence of a thick layer (179 nm) with a high density of amine. Upon retraction of the AFM tip, the rupture length is elongated up to 15.1 nm (Figure 6(b’), Table 2) until pulling off. On the other hand, the adhesion force of the PEI-grafted PMMA was measured at 23.9 nN (Figure 6(c), Table 3), which is quite small compared to the thick PEI layer on the PEI-coated PMMA (Figure 6(b), Table 3). The PEI layer of the PEI-grafted PMMA is thin (<1 nm), as obtained by XPS analysis. After the extraction process, the unbounded PEIs were removed from the PMMA substrate. Therefore, the covalently bounded PEI molecules remained as a monolayer on PMMA. Typically, the thin layer is expected to be less adhesive to the AFM tip than the thicker ones [36], and the interaction of the tip is considered to be not with the surface of the thin PEI, but with the surface of the PMMA substrate.

Nevertheless, the behavior of the retraction curve was observed to be not simple after passing the maximum adhesion force. In Figure 6, the detachment process of PEI-grafted PMMA shows that the retardation of the adhesion is composed of two steps (I and II). The first step is similar to the slope of the retraction curve from others, while the second step is moderate, resulting in a longer rupture length than PMMA (Figure 6(c’), Table 3). This phenomenon can be explained by the fact that the pulling off is decelerated because of the attachment of the PEI molecule bonded on PMMA to the AFM tip by chemical or physical interactions.

#### 3.1.3. Contact Angle Measurements

Wettability is one of the most important characteristics for surface modification and it is easily measured by observing the shape of the water droplet. Figure 7b,c show surface wettability with water remaining after rinsing on the PMMA with and without a 750 K PEI layer after laser irradiation, respectively. The extraction process with EtOH and water was applied to both of the samples. The presence of residual water only on the laser-irradiated area is evidence that the surface grafted by PEI became hydrophilic (Figure 7c). Generally, the PEI molecule is considered a hydrophilic attribute because it has plenty of amines. In contrast, the surface without PEI molecules showed a hydrophobic property and was not clearly distinguished between irradiated and non-irradiated areas (Figure 7b). The wettability of the surface by water on bare PMMA is not influenced by laser treatment.

The dynamic contact angles were measured to confirm the surface wettability. The contact angle of PMMA has been reported around 70~90° for water by Kwok et al. [37]. In this paper, the contact angle for the bare PMMA was 81.7°, which is almost hydrophobic, close to 90° in case of advancing angle, *θ*_a_, measured with water (Table 4). However, the receding angle, *θ*_r_, was approximately 50.4°. The contact angle hysteresis, *∆θ*, defined as the difference between the advancing and the receding angles, is due to deviations of the surface roughness, chemical heterogeneities [38] of the surface, molecular reorientations, drop sizes [39], and so on. The *∆θ* of bare PMMA and PEI-grafted PMMA were calculated to be approximately 31.3° and 29.8°, which are quite similar. Based on this, we assumed that the deviations of the surface characteristics were not significantly different between bare PMMA and PEI-grafted PMMA; therefore, the hysteresis was not studied in this work. Nevertheless, both advancing and receding contact angles were changed drastically after PEI modification, with −24.6° and −23.1° for *θ*_a_ and *θ*_r_, respectively. Generally, *θ*_a_ of the PEI surface is reported to be around 40 ~ 60° [40], which agrees well with the result of the PEI-grafted PMMA in this study. It was observed that *θ*_r_ decreased further for PEI-grafted PMMA when measured with 0.1 M of HCl. We assume that amine cations (-NH_3_^+^) were produced in the HCl solution. Thus, the receding angle was decreased by the effect of ionic strength.

#### 3.1.4. Zeta Potential Measurements

Zeta potential is important to understand the surface charges for various Coulombic interactions, such as with chemicals, DNA, nanoparticles, carbon nanotubes, dyes, metal ions, and so on. The electrokinetic measurements can give information on the presence of positive or negative charges on the surface and on the main driving force of the reaction with dye and metal ions as a function of the sorption conditions.

The surface charges of the bare PMMA and PEI-grafted PMMA were analyzed by zeta potential measurement as a function of the pH at room temperature. Figure 8 shows the pH dependence of the zeta potential of bare PMMA in the range of pH 3 to 10. Upon increasing pH, the zeta potentials tend to decrease, passing the isoelectric points (IEP), which are the pH values when the zeta potential is zero, and then reach negative values. The surface of bare PMMA is acidic because the IEP is at pH 3.6 and the surface charge is around −50 mV at pH 7. It is well explained that the ester group in PMMA, having partially negative oxygen of the carbonyl group, forms hydrogen bonds in water. The zeta potential of bare PMMA was around five times higher at pH 10 because the hydroxide adsorption occurs especially at the hydrophobic PMMA interface rather than hydrophilic PEI-grafted PMMA [41]. The zeta potential of bare PMMA is decreased linearly as a function of the pH and it reaches the stable state in the plateau phase at approximately −57 mV in the range of pH 8 to 10. 

As seen in Figure 8, the zeta potential of PEI-grafted PMMA is distinguished from that of PMMA. The IEP of the PEI-grafted PMMA is approximately 6.7. This shift in the IEP indicates that PEI-grafted PMMA has a more cationic surface than bare PMMA. The charge is positive in the range from 3 to 6.7. Moreover, the zeta potential (28 mV) of PEI-grafted PMMA at pH 3 is more than twice that of bare PMMA (11 mV) because of the cationic -NH_3_^+^ groups. With increased pH, the zeta potential of PEI-grafted PMMA decreases due to the transformation of protonated amines to neutral ones. The decrease in zeta potential continues and becomes negative when the interface adsorbs more hydroxyl ions in the basic environment, resulting in a negative surface charge at pH higher than 6.7. As we can see in the zeta potential measurements, a significant change in surface charges after PEI modification could be observed.

### 3.2. Reaction with Dye and Metal

The PEI molecules were covalently bonded with amide anchors on the PMMA substrate discussed in the XPS study. As a result, NH_2_-terminated PMMA was achieved using CO_2_ laser treatment. To investigate the selective reaction of the functionalized PMMA by laser irradiation, Eosin Y molecules and Copper (ΙΙ) acetate were used for the ionic reaction and metal chelation, respectively. In this study, 750K PEI was used because of its high amine density to improve the analytic signal. 

#### 3.2.1. Eosin Y Binding

In the previous section regarding the results of the investigation of the effect of energy density, Eosin Y was applied as a strategy to visualize the results after laser irradiation. At neutral pH, the anionic Eosin Y reacts with protonated cationic PEI attached to the PMMA surface by electrostatic attraction (forming an Eosin Y–PEI complex). The reacted area is easily recognized by a change in color from orange to pink because of the maximum absorption (λ_max_) shift (forming the Eosin Y–PEI complex). The absorption spectrum of the Eosin Y–PEI complex is shown in Figure 9. As shown in Figure 9(a), the bare PMMA substrate has no absorption in a UV–Vis wavelength. While the Eosin Y molecules with chromophores show one band with λ_max_ at 520 nm, the absorption spectrum of the film coated on PMMA splits into two bands at 491 and 528 nm (Figure 9(b)). In an aqueous solution, Eosin Y is mostly in a monomer form with one λ_max_. However, a blue-shifted absorption can be observed around 490 nm according to the analysis of Yoshida [42] in the case of the film of Eosin Y. It is reported that the intermolecular attraction between chromophores leads to the formation of assemblies due to the close packing [43], and it is mostly dimer form, resulting in the shoulder near 490 nm.

After Eosin Y–PEI complex formation, the λ_max_ shifted from 528 to 537 nm (Figure 9(c)). PEI has a large number of amino groups, which play a role as electron donors. Meanwhile, Eosin Y, a xanthene dye with redox capability, can act as an electron acceptor [44,45]. Therefore, the photoinduced electron transfer between Eosin Y and PEI is the reason for the red shift of λ_max_. In this case, there are mixtures of Eosin monomer, dimer, Eosin Y–PEI complex, and so on. However, the λ_max_ shifts slightly more to the red (541 nm, Figure 9(d)) if the Eosin Y binds to the PEI-grafted PMMA that is irradiated by the CO_2_ laser, and the absorption of the shoulder from the Eosin dimer near 500 nm decreases significantly by removing the physisorption of molecules from the PMMA surface. This spectrum is attributed only to the Eosin Y complex with PEI grafted on the PMMA.

Notably, the Eosin Y that reacted with unbounded PEI was removed completely by the extraction procedure, because it exhibited the same absorption spectrum as bare PMMA in the UV–Vis spectrum (Figure 9(e)). Consequently, the physically adsorbed Eosin Y–PEI complex on PMMA was easily dissolved by ethanol and water rinsing. This result confirms that the extraction process developed in this study is satisfactory for the selective modification with functional groups on a plastic substrate. Furthermore, the indirect screening test using dye is one of the suitable methods to visualize the result quickly to confirm whether the surface is modified or not. 

#### 3.2.2. Metal Complex

A selective metallization on a plastic substrate is one of the most interesting studies for various applications. PEI is a typical polymeric amine having a chelation complex with various metal ions by coordinating bonds. Among them, it is well known that Cu^2+^ forms a stable complex having a four-coordinate planar structure with PEI [46]. In this paper, PEI-grafted PMMA was used as a ligand with Copper (ΙΙ) acetate for chemical metallization. The PEI-coated PMMA was irradiated for PEI grafting in an area with 6 W of laser power and 305 mm/s of scan speed at 20 mm out of focus, and then the sample was rinsed with D.I. water and ethanol several times. Then, the PEI-grafted area was fabricated in the laser-irradiated area, and the non-irradiated area remained as bare PMMA. After metal chelation of the PEI-grafted PMMA substrate in the saturated Copper (ΙΙ) acetate solution, the chemical components of the surface were analyzed using energy-dispersive X-ray spectrometry (EDS) in order to compare the binding of the Cu complex in the laser-irradiated area and non-irradiated area. Figure 10 shows the X-ray spectra of the Cu–PEI complex on the irradiated area (solid line) and non-irradiated area (dotted line). The carbon, nitrogen, and oxygen peaks are around 0.28, 0.39, and 0.52 keV, respectively. At around 0.93 keV, the peak corresponding to Cu L_α_ was only detected in the area of irradiation. The nitrogen signal in the spectrum was also presented only in the laser-irradiated area. EDS quantitative results of all elements are summarized in Table 5. After laser irradiation, PEI molecules were chemically bound on PMMA and reacted with copper ions, forming the Cu–PEI complex. As a result, the carbon contents were decreased from 77.8 (corresponds to bare PMMA) to 69.7 at %. The oxygen contents in both areas were almost identical, which means that the chemical composition of the irradiated area is influenced by the PMMA substrate. This is one of the reasons for the nitrogen concentration (9.8 at% in Table 5), which is less than the theoretical value (33% of N in PEI). The Cu concentration is approximately 0.5 at %, which is influenced by the pH of the PEI solution. The number of free amino groups is strongly related to the pH of the solution, with PEI being protonated at low pH. M. Amara and H. Kerdjoudj studied the kinetics for metal–PEI complexes and reported that amino groups were protonated at 64%, 50%, and 32%, at pH 4, 8.8, and 9.0, respectively [47,48]. These protonated amine ions change the metal–PEI complex form. In this study, the pH value of saturated Copper (ΙΙ) acetate solution was approximately 5.5, which is low enough to protonate amine groups to more than 50%. To increase the Cu concentration for Cu–PEI chelation, it is recommended to be performed at high pH. Even with a low concentration of Cu, it was selectively chelated on the PEI-grafted PMMA, which was irradiated by a CO_2_ laser, as shown in the EDS analysis. This is notable for future investigations such as electroless metal plating to nucleate a metal catalyst on a PEI-grafted patterned plastic substrate.

## 4. Conclusions

Poly(ethyleneimine) (PEI) is chemically grafted on poly(methyl methacrylate) (PMMA) by irradiation of a conventional CO_2_ laser. Effective photothermal energy for the chemical reaction was investigated by controlling the laser parameters, such as power, scan rate, and focal length. Finally, the amino groups were modified with amide bonds on the PMMA surface, and the chemical bonds were characterized by XPS and TOF-SIMS. The hydrophobic PMMA surface changed to hydrophilic, showing a drastic decrease in dynamic contact angles after PEI modification. Furthermore, the surface charge of PEI-grafted PMMA became more cationic than bare PMMA because of the cationic -NH3+ groups. Adhesion force, which is another surface property proving the surface change after PEI modification, was studied by measuring the force–distance curves between the AFM tip and PEI-modified surface and explained chemical or physical interactions well. By laser irradiation, PEI can be modified on the PMMA surface with high selectivity after the extraction process, which removes unbounded PEI molecules from the PMMA surface. To observe the successful PEI modification, a dye (Eosin Y) was applied and we visualized the modified surface by the color change of the Eosin Y–PEI complex. The PEI molecules that were chemically bound on PMMA reacted with copper ions, forming Cu–PEI complexes. These results support the notion that chemical modification by laser irradiation is one of the most promising methods for different materials in various applications.

## Figures and Tables

**Figure 1 polymers-14-02041-f001:**
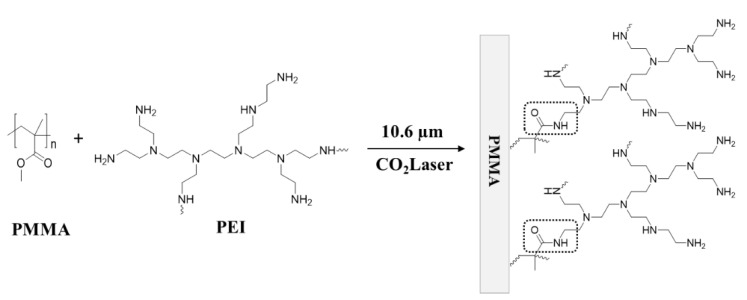
Schematic drawing of aminolysis reaction for laser-assisted PEI grafting on PMMA.

**Figure 2 polymers-14-02041-f002:**
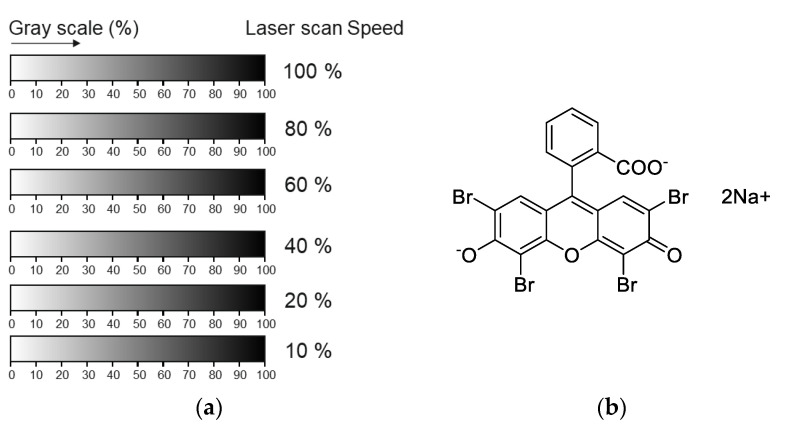
(**a**) Drawing for the laser experiments, and (**b**) the molecular structure of Eosin Y.

**Figure 3 polymers-14-02041-f003:**
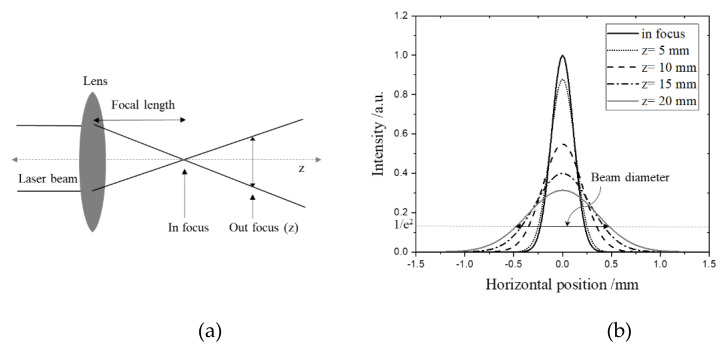
Laser beam profiles according to the Z positions. (**a**) Schematic drawing for the laser beam divergence; (**b**) simulated Gaussian intensity distributions according to the Z position.

**Figure 4 polymers-14-02041-f004:**
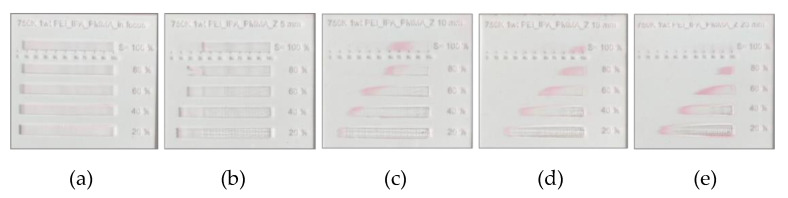
Investigation into the optimum conditions of chemical reactions with different laser power and scan speed according to the Z positions. Experimental results after Eosin Y binding in different Z positions (**a**) in focus, (**b**) Z = 5 mm, (**c**) Z = 10 mm, (**d**) Z = 15 mm, and (**e**) Z = 20 mm, respectively. Gray wedge according to Experimental Section.

**Figure 5 polymers-14-02041-f005:**
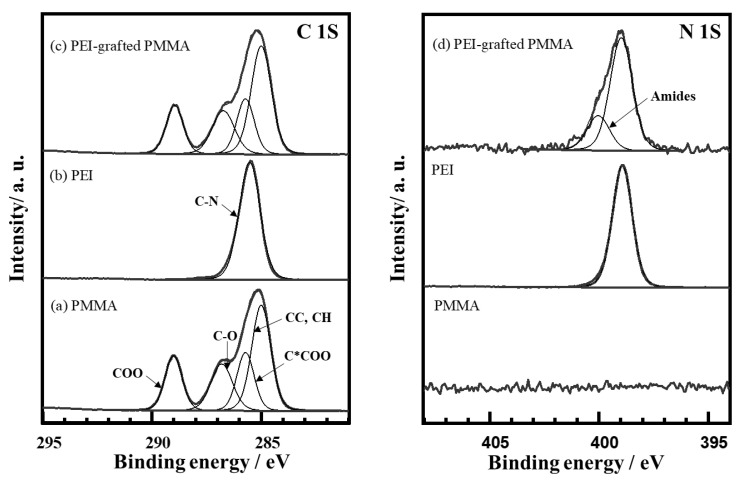
C 1s and N 1s analysis measured by XPS for bare PMMA, PEI-coated PMMA, and PEI-grafted PMMA.

**Figure 6 polymers-14-02041-f006:**
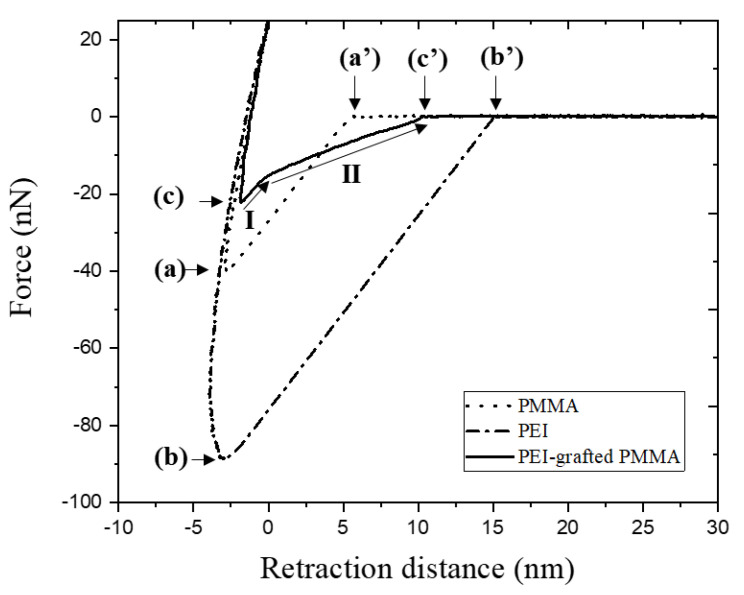
Force–distance plots of AFM measurements (retraction curves) for bare PMMA, PEI-coated PMMA, and PEI-grafted PMMA.

**Figure 7 polymers-14-02041-f007:**
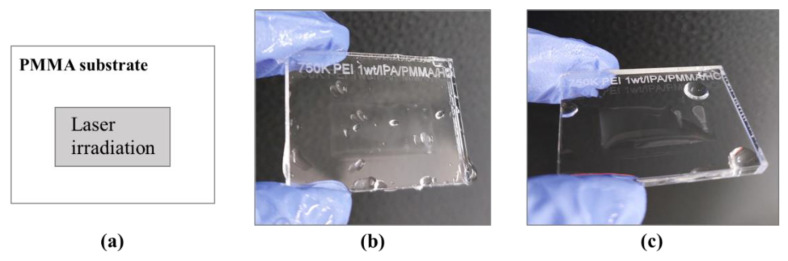
Surface wettability test. The schematic of the sample (**a**) and pictures after rinsing with water on the laser-treated bare PMMA (**b**) and PEI-grafted PMMA (**c**).

**Figure 8 polymers-14-02041-f008:**
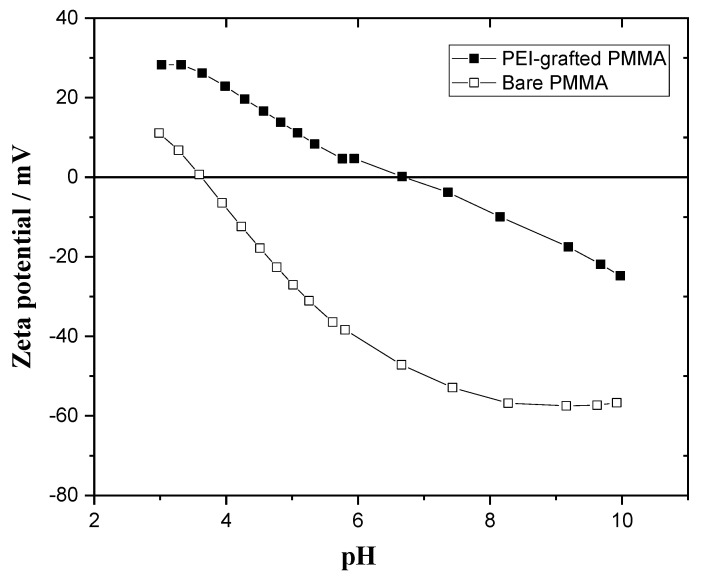
Zeta potential versus pH of bare PMMA and PEI-grafted PMMA.

**Figure 9 polymers-14-02041-f009:**
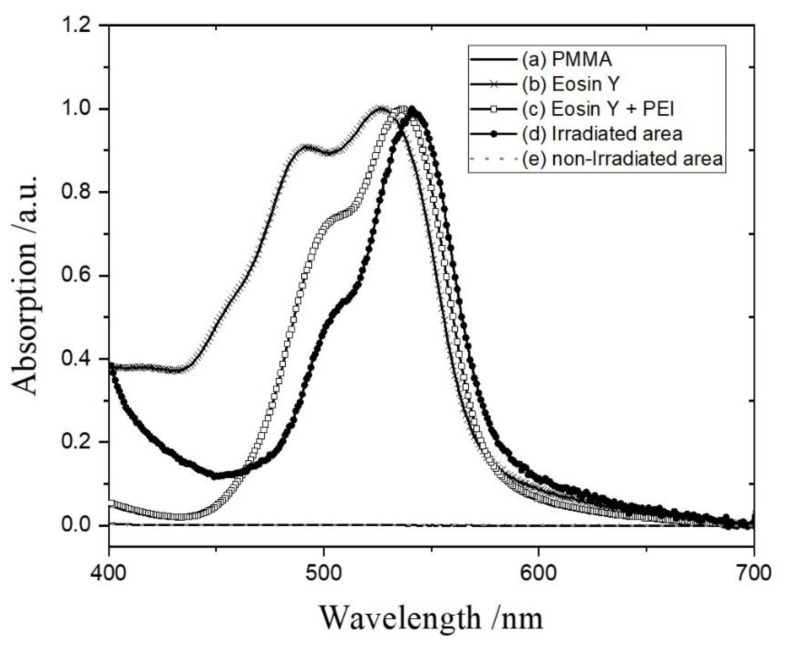
UV–Vis spectra of bare PMMA (a), Eosin Y cast on PMMA (b), Eosin Y reacted with PEI-cast PMMA (c), Eosin Y–PEI complex obtained by laser irradiation after extraction (d), and non-irradiated area (e) after extraction.

**Figure 10 polymers-14-02041-f010:**
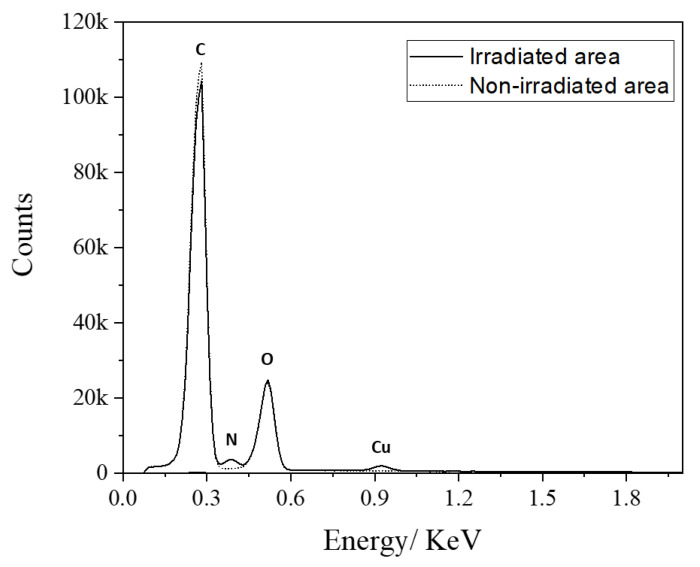
X-ray spectra of the Cu–PEI complex on the irradiated area (solid line) and non-irradiated area (dotted line).

**Table 1 polymers-14-02041-t001:** Surface elemental compositions.

Sample	C	O	N
PMMA	79.4	20.6	-
PEI-coated PMMA	72.6	1.8	25.5
PEI-grafted PMMA	79.4	17.5	2.9

**Table 2 polymers-14-02041-t002:** Fragment intensities of TOF-SIMS for PEI-coated PMMA and PEI-grafted PMMA normalized to the total ion intensity.

Fragments	Mass (u)	PEI-Coated PMMA	PEI-Grafted PMMA
CN^−^	26.01	6.49 × 10^−2^	8.97 × 10^−3^
CNO^−^	42.01	2.55 × 10^−2^	3.98 × 10^−3^
NH^−^	15.01	3.68 × 10^−3^	4.58 × 10^−4^

**Table 3 polymers-14-02041-t003:** Adhesion force and rupture length for bare PMMA, PEI, and PEI-grafted PMMA ^a^.

Samples	Adhesion Force, nN	Rupture Length, nm
PMMA	(a) 40.8 ± 4.8	(a’) 6.1 ± 0.4
PEI-coated PMMA	(b) 87.9 ± 6.1	(b’) 15.1± 1.4
PEI-grafted PMMA	(c) 23.9 ± 3.3	(c’) 9.7 ± 1.3

^a^ All the values are average of 9 data and shown with standard deviations.

**Table 4 polymers-14-02041-t004:** Contact angles for laser-treated bare PMMA and PEI-grafted PMMA ^a^.

Samples	with H_2_O			with 0.1 M HCl		
	*θ* _a_	*θ* _r_	∆*θ*	*θ* _a_	*θ* _r_	∆*θ*
Laser-treated bare PMMA	81.7	50.4	31.3	80.8	55.1	25.7
PEI-grafted PMMA	57.1	27.3	29.8	53.4	19.9	33.5

^a^ All the contact angle values are in deg.

**Table 5 polymers-14-02041-t005:** EDS analysis ^a^ results of Cu–PEI complex.

	Cu–PEI Complex			
	on the Irradiated Area		on the Non-Irradiated Area	
Element	At %	Wt %	At %	Wt %
C	69.7	63.1	77.8	72.6
N	9.8	10.3	1.4	1.5
O	20.1	24.2	20.9	25.9
Cu	0.5	2.4	0	0

^a^ Chemical analysis was obtained in the area of 75 μm × 40 μm for each sample.

## Data Availability

Not applicable.

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
