# Peer review of "Laser-Assisted Direct Grafting of Poly(ethyleneimine) on Poly(methyl methacrylate)"

_polymers, 2022, doi:10.3390/polym14102041_

Round 1

Reviewer 1 Report

The paper by Park et al., on "Laser-assisted direct grafting of Poly (ethyleneimine) on poly 2(methyl methacrylate)" is  very interesting field of research to modify by surfaces with biocompatable polymers. Authors have carried the process and the analysis well and the design of the experiment is good.

  1. I am interested in atom economy after laser applications, since polymer is PEI with imine groups, what is the total conversion rate.
  2. Poly-ethyleneimine (PEI) with a high density of amino groups is chemically grafted on Poly-methyl-methacrylate (PMMA) by irradiation of CO2 laser (10.6 m). Laser parameters such as power, scan rate, and focal length are observed to play an important role in order to introduce effective photothermal energy for the chemical reaction. However, the substrate effect are not considered. 
  3. The film thickness has been calculated from the AFM image by measuring the gap between the highest and lowest. I am not sure how accurate that is, but it might have variation and then standard deviation needs to be calculated with nearest error correction. 
  4. Thickness after wash, does it effect the density or the wetabilty of the film.

Reviewer 2 Report

The authors investigated the grafting of PEI to PMMA by means of the laser process. The manuscript has merit for publication. Some suggestions are reported below to improve the manuscript:

1°) Please explain better in the introduction the novelty of the manuscript. In addition, the authors should add a specific review of laser modified polymers.

2°) Please add the PEI and PMMA commercial code; the density; the fluidity index.

3°) Page 2. What were the solvents? Add in that part of materials;

4°) Page 3. Were the parameters adopted in the laser experiment chosen randomly?

5°) Did the authors perform Fourier Transform Infrared Spectroscopy (FTIR) analysis? It is important to check the emergence of new bands that have been grafted.

5°) Authors should investigate chemical modification by nuclear magnetic resonance (NMR). NMR chemical shifts should be used.

Round 2

Reviewer 2 Report

Dear Editor,

The authors answered the questions satisfactorily. Improvements have been added to the manuscript, and therefore have merit for publication.

Yours sincerely